# CHARGED POINT NORMALIZATION

# AN EFFICIENT SOLUTION TO THE SADDLE POINT PROBLEM

**Armen Aghajanyan**
Bellevue, WA 98007, USA
`armen.ag@live.com`

## ABSTRACT

Recently, the problem of local minima in very high dimensional non-convex optimization has been challenged and the problem of saddle points has been introduced. This paper introduces a dynamic type of normalization that forces the system to escape saddle points. Unlike other saddle point escaping algorithms, second order information is not utilized, and the system can be trained with an arbitrary gradient descent learner. The system drastically improves learning in a range of deep neural networks on various data-sets in comparison to non-CPN neural networks.

## 1 SADDLE POINT PROBLEM

### 1.1 INTRODUCTION

Recently more and more attention has focused on the problem of saddle points in very high dimensional non-convex optimization. Saddle points represent points in the optimization problem where the first order gradients are all zero, but the stationary point is neither a maxima or a minima. The saddle point of a function can be confirmed by using the eigenvalues of Hessian matrix. If the set of eigenvalues contains at least one negative eigenvalue and at least one positive eigenvalue the point is said to be a saddle point. One way to analyze the prevalence of saddle point is to assign a joint probability density to the eigenvalues of the Hessian matrix at a critical point.

- If the eigenvalues are all negative, then the critical point is a local maximum
- If the eigenvalues are all positive, then the critical point is a local minimum.
- If the eigenvalues contain at least one positive and at least one negative eigenvalue then the point is a saddle point.

If $p(\lambda_1, \lambda_2, ..., \lambda_n)$ is the joint probability density function then the probability that the Hessian matrix resembles a saddle point is, given that the Hessian is not singular is:

$$
\begin{aligned}
&1 - \int_0^\infty \int_0^\infty ... \int_0^\infty p(\lambda_1, \lambda_2, ..., \lambda_n) d\lambda_1 d\lambda_2 ... d\lambda_n \\
&- \int_{-\infty}^0 \int_{-\infty}^0 ... \int_{-\infty}^0 p(\lambda_1, \lambda_2, ..., \lambda_n) d\lambda_1 d\lambda_2 ... d\lambda_n
\end{aligned}
\tag{1}
$$

Another way to interpret the expression above is to realize that each of the two $n$-integrals represents the joint density summation of the two hyper-cubes, one in the direction of all the positive axis, and the other in all the negative axis. Each respectively representing minimas and maximas.

**Theorem 1** *The space of eigenvalues of a non-singular Hessian matrix that represent minimas and maximas in comparison to the total space, decreases by $2^n$ asymptotically.*

The amount of unique hypercubes starting from the origin and spanning along the axis is $2^n$. The amount of hypercubes representing minimas and maximas is two. Therefore the fraction of the space that contains the eigenvalues that would indicate either a minima or a maxima is $2^{1-n}$, where $n$ represents the dimensionality of the Hessian matrix.

What this shows is that as we increase the dimensionality of our optimization problem, the fraction of the total space that represents either a local minima or maxima decreases exponentially by a factor of two.

Although this interpretation gives some intuition behind the saddle point problem, we cannot conclusively say that the probability of a critical point being a saddle point increases exponentially because we do not know the behavior of the joint probability function.

## 1.2 GRADIENT DESCENT BEHAVIOR AROUND SADDLE POINTS

To understand the shortcomings of first order gradient descent algorithms around saddle points we will analyze the neighborhood a saddle point. Given a function $f$, the Taylor expansion around the saddle point $x$ is given by:

$$f(\boldsymbol{x} + \boldsymbol{\delta}) = f(\boldsymbol{x}) + \frac{1}{2}\boldsymbol{\delta}^T \boldsymbol{H} \boldsymbol{\delta} \tag{2}$$

The first order term disappears because we are at a critical point. Denoting $e_1, e_2, ..., e_n$ as the eigenvectors of the non-degenerate Hessian $H$, and $\lambda_1, \lambda_2, ..., \lambda_n$ as the respective eigenvalues, we can use the change of coordinates methods to rewrite the Taylor expansion in terms of the eigenvectors:

$$\boldsymbol{v} = \frac{1}{2} \begin{pmatrix} \boldsymbol{e}_1^T \\ ... \\ \boldsymbol{e}_n^T \end{pmatrix} \boldsymbol{\delta}$$

$$f(\boldsymbol{x} + \boldsymbol{\delta}) = f(\boldsymbol{x}) + \frac{1}{2}\sum_{i=1}^{n} \lambda_i (\boldsymbol{e}_i^T \boldsymbol{\delta})^2 = f(\boldsymbol{x}) + \sum_{i=1}^{n} \lambda_i \boldsymbol{v}_i^2 \tag{3}$$

From the last equation we can analyze the behavior of first order gradient descent algorithms. Specifically by looking at the behavior with respect to the signs of the eigenvalues. If eigenvalue $\lambda_i$ is positive then the optimization point will move toward the critical point $\boldsymbol{x}$. If eigenvalue $\lambda_i$ is negative the optimization point will move away from the critical point.

This shows that the direction of the gradient descent algorithm is not the problem with gradient descent algorithms around saddle points, but rather the step of the algorithm. This problem is sometimes amplified because of the plateaus surrounding the critical point, as shown in (Saad & Solla, 1996). Another complication visible from equation 2, is that if the step size is greater than $\max \lambda^{-1}$, the gradient descent algorithm will begin to diverge. Therefore one large eigenvalues of the surface of the error function, will force the gradient descent algorithms to take very small steps in all the other directions.

A very similiar derivation and explanation was shows in (Dauphin et al., 2014)

## 2 CHARGED POINT NORMALIZATION

### 2.1 METAPHOR

The metaphor for our method goes as follows. The current point in our optimization is a small positively charged point that is moving on the neutral surface of error. Our normalization works by dynamically placing other positively charged points around the surface of error to 'push' our optimization point away from undesirable positions. Optimally we would run the gradient descent algorithm until convergence, check if the converged point is a saddle point, place a positively charged point near the saddle point and continue the optimization. This metaphor was what gave inspiration to the derivation of our normalization.

## 2.2 INTRODUCTION

The general optimization problem is defined as:

$$\mathcal{L}(f; \boldsymbol{X}, \boldsymbol{Y}) = \sum_{i=1}^{n} V(f(\boldsymbol{X}_i), \boldsymbol{Y}_i) + \lambda R(f) \tag{4}$$

The formulation is static, given the same function and the same $\boldsymbol{X}$ and $\boldsymbol{Y}$ the loss will always be equal. Our formulation introduces a dynamic normalization function $R$. Therefore the loss function becomes defined as:

$$\mathcal{L}_t(f; \boldsymbol{X}, \boldsymbol{Y}) = \sum_{i=1}^{n} V(f(\boldsymbol{X}_i), \boldsymbol{Y}_i) + \lambda R_t(f) \tag{5}$$

The function $f$ contains dynamic parameters $\boldsymbol{W}_1^t, \boldsymbol{W}_2^t, ..., \boldsymbol{W}_n^t$, while the function $R$ contains parameters: $\beta$, $p$, $\phi$ and $\hat{\boldsymbol{W}}_1^t, \hat{\boldsymbol{W}}_2^t, ..., \hat{\boldsymbol{W}}_n^t$, symbolizing the decay factor, norm, merge function and merge values respectfully. The $t$ term in $\boldsymbol{W}_n^t$ represents the value of $\boldsymbol{W}_n$ at time $t$ of the optimization algorithm. Charged Point Normalization is now defined as:

$$R_t(f) = \frac{e^{-\beta t}}{\sum_{i=1}^{n} ||\boldsymbol{W}_i^t - \hat{\boldsymbol{W}}_i^t||_p} \tag{6}$$

The update for the merge values is defined as:

$$\hat{\boldsymbol{W}}_i^{t+1} = \phi(\hat{\boldsymbol{W}}_i^t, \boldsymbol{W}_i^t)$$
$$\hat{\boldsymbol{W}}_i^1 = \boldsymbol{W}_i^1 + \boldsymbol{\epsilon} \tag{7}$$

where $\epsilon$ is a source of random error to ensure we do not have a division by zero. In our experiments $\epsilon$ was a matrix of the same size as $W_i^1$ with random entries sampled from a normal distribution with a zero mean and a very small standard deviation.

What this type of normalization attempts to do is to reward the optimization algorithm for taking steps that maximize the distance between the new point and the trailing point. This can be seen as a more dynamic and adaptive version of momentum that kicks in when the optimization problem settles down into a saddle point or long plateau. That being said, CPN can still be used with traditional momentum methods, as shown by the experiments below.

## 2.3 CHOICE OF HYPERPARAMETER

The $\phi$ function can be any function that merges the two parameters into one parameter of the same dimension. Throughout this whole paper we used the exponential moving average for our $\phi$ function.

$$\phi(\hat{\boldsymbol{W}}_i^t, \boldsymbol{W}_i^t) = \alpha \hat{\boldsymbol{W}}^t + (1 - \alpha) \boldsymbol{W}_i^t$$
$$\alpha \in (0, 1) \tag{8}$$

Although to keep up with the metaphor Coulomb's inverse squared law did not work as well as projected, through trial and error, the $p$ value that worked the best was 1. The 1-norm simply is the sum of absolute values.

## 3 EXPERIMENTS

### 3.1 INTRODUCTION

Charged Point Normalization was implemented in Theano (Bastien et al., 2012) and integrated with the Keras (Chollet, 2015) library. We utilized the convolutional neural networks and recurrent networks in the keras library as well. All training and testing was run on a Nvidia GTX 980 GPU. We

do not show results on a validation set, because we care about the efficiency and performance of the optimization algorithm, not whether or not it overfits. The over-fitting of a model is not the fault of the optimization routine but rather the fault of the field it is optimizing over. All comparisons between the standard and charged model, started with identical set's of weights. Throughout all of our experiments, we utilize a softmax layer as the final layer, and consequently all the losses measured throughout this paper will be in terms of categorical cross-entropy. We used the train split of each data-set.

## 3.2 SIMPLE DEEP NEURAL NETWORKS

### 3.2.1 MNIST: MULTILAYER PERCEPTRON

The first test conducted was using a multilayer perceptron on the MNIST dataset. The architecture of the neural net contained layers with sizes $784 \rightarrow 512 \rightarrow 512 \rightarrow 10$. All intermediate layers contained rectified linear activations (He et al., 2015), while the final layer is a softmax layer. Between layers, dropout (Hinton et al., 2012) with a probability of 0.2 was added. We compare the standard batch gradient descent algorithm with a step size of 0.001 and batchsize of 128, on the net described above and the same net with Charged Point Normalization (CPN). The CPN hyper-parameters were: $\beta = 0.001$, $\lambda = 0.1$ with the moving average parameter $\alpha = 0.95$. The loss we were optimizing over was categorical cross entropy.

### 3.2.2 MNIST:DEEP AUTOENCODER

The second test conducted on simple neural networks, was in the form of an autoencoder. The architecture of the autoencoder contained layers with sizes $784 \rightarrow 512 \rightarrow 512 \rightarrow 10 \rightarrow 512 \rightarrow 512 \rightarrow 10$. All layers contained rectified linear activations. Between layers, dropout with a probability of 0.2 was added. The set up of the experiment is almost identical to the previous experiment. The only difference being that in this case, we optimized for binary cross-entropy.

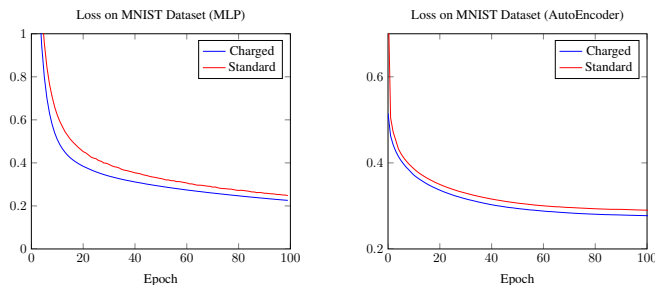

### 3.2.3 NOTES

It is interesting to note that when the optimization problem is relatively simple, more specifically if the optimization algorithm takes smooth steps, CPN allows the optimization algorithm to take bigger steps in the correct direction. CPN does not display any periodic or chaotic behavior in this scenario. This is not the case for more complicated optimization problems that will be presented below.

## 3.3 CONVOLUTIONAL NEURAL NETWORKS

### 3.3.1 CIFAR10

The next experiment conducted was using a convolutional neural network on the CIFAR10 (Krizhevsky et al., a). The architecture was as such:

Convolution2D (32,3,3) $\rightarrow$ ReLU $\rightarrow$ Convolution2D (32,3,3) $\rightarrow$ ReLU $\rightarrow$ MaxPooling (2,2) $\rightarrow$ Dropout (0.25) $\rightarrow$ Convolution2D (64,3,3) $\rightarrow$ ReLU $\rightarrow$ Convolution2D (64,3,3) $\rightarrow$ ReLU $\rightarrow$ MaxPooling (2,2) $\rightarrow$ Dropout (0.25) $\rightarrow$ Dense (512) $\rightarrow$ ReLU $\rightarrow$ Dropout (0.5) $\rightarrow$ Dense (10) $\rightarrow$ Softmax

Convolution2D takes the parameters, number of filters, width and height respectfully. Dense take one parameter describing the size of the layer. MaxPooling takes two parameters that signify the

pool size. ReLU is the rectified linear function, while Softmax is the softmax activation function. The optimization algorithm used was stochastic gradient descent with a learning rate of $0.01$, decay of $1e-6$, momentum of $0.9$, with nesterov acceleration. The batch size used was 32. The hyper-parameters for CPN were: $\beta = 0.01$, $\lambda = 0.1$ with the moving average parameter $\alpha = 0.95$. 10,000 random images were used from the CIFAR10 data-set instead of the full dataset to speed up learning.

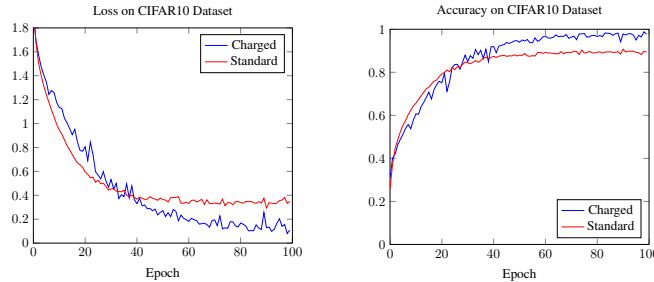

It is interesting to note that CPN performs worse until the optimization algorithm reaches the 'elbow' of the curve, where then CPN continues along its path, while the standard algorithm begins to converge. CPN also takes steps that are much less 'optimal' in the greedy sense, which is why both the loss and accuracy curve behave partially chaotic.

### 3.3.2 CIFAR100

The CIFAR100 (Krizhevsky et al., b) setup was nearly identical to the CIFAR10 setup. The same architecture of the neural network was used. The only difference was in the $\lambda$ parameter in the normalization term, which in this case was equal to $0.01$. $20,000$ random images were used.

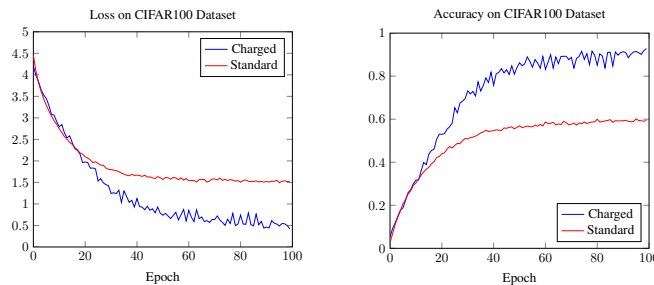

The same behavior as in the CIFAR10 experiment was exhibited. The elbow of the loss curve was the point where CPN began to outperform standard optimization.

## 3.4 RECURRENT NEURAL NETWORKS

### 3.4.1 INTRODUCTION

Recurrent neural networks are notorious for being hard to train, and having a tendency to generally underfit (Pascanu et al., 2012).
In this section we show that CPN successfully escapes saddle points presented in recurrent neural networks.

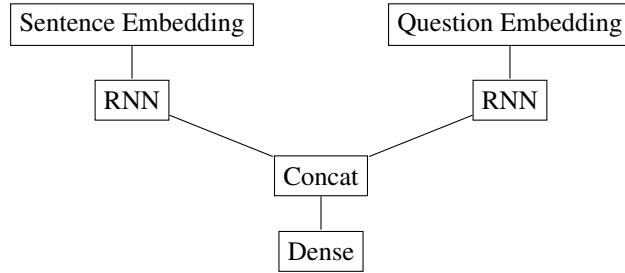

Figure 1: Architecture for BABI Test

### 3.4.2 BABI

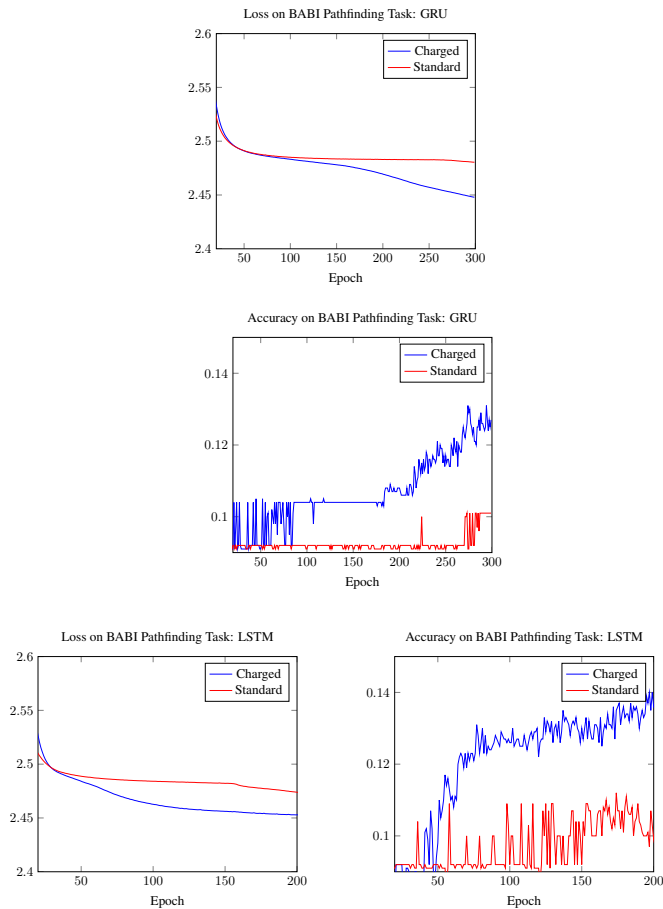

We selected the path-finding problem of the BABI dataset due to it being the most difficult task. The architecture consisted of two recurrent networks and one standard neural network. Each of the recurrent neural networks had a structure: Embedding → RNN. The embedding, sentence and query hidden layer size was set to 3. The final network concatenated the two recurrent network outputs and fed the result into a dense layer with an output size of vocabsize. Refer to figure 1 for a diagram. We ran our experiment with two different recurrent neural network structures: Gated Recurrent Units (GRU) (Chung et al., 2014) and Long Short Term Memory (LSTM) (Hochreiter & Schmidhuber, 1997) . The ADAM (Kingma & Ba, 2014) optimization algorithm was used for both recurrent structures with the parameters: $\alpha = 0.001$, $\beta_1 = 0.9$, $\beta_2 = 0.999$, $\epsilon = 1e - 08$. For the LSTM architecture, CPN hyper-parameters were: $\beta = 0.0025$, $\lambda = .03$, $\alpha = 0.95$. For the GRU architecture, CPN hyper-parameters were: $\beta = 0.1$, $\lambda = .1$, $\alpha = 0.95$.

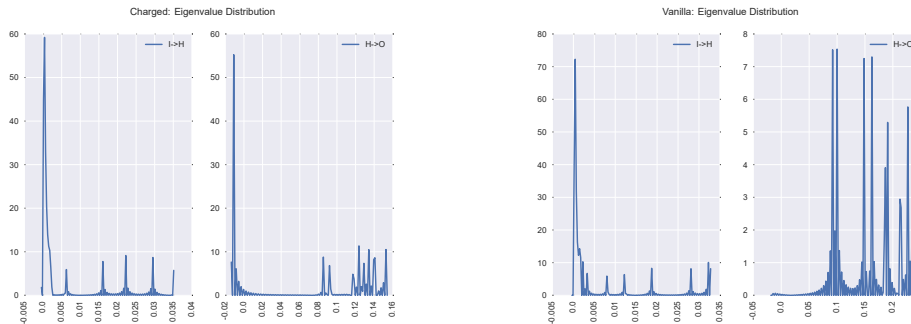

From the accuracy graphs we can see the CPN causes the recurrent network to escape the saddle point earlier than a recurrent network with no CPN.

## 4 Normalization Behavior

### 4.1 Exploration vs Exploitation

In a standard gradient descent with no normalization, the updates taken by the algorithm are always greedy, in terms of always minimizing the loss of the model. Their is no exploration done; gradient descent is by nature a greedy algorithm, optimizing only locally. What CPN allows the gradient descent to do, is to move in non-optimal directions early on in the optimization routine, while still allowing for precise finetuning at the end of the model. This trade-off is controlled by the $\beta$ parameter.

### 4.2 Behavior Around Saddle Points

A vanilla neural network with one single hidden layer was trained on a down sampled $8 \times 8$ version of the MNIST dataset (Lecun et al., 1998). Full gradient descent was ran on the $10,000$ random images until convergence. We compare the differences between the eigenvalue distributions between the neural network with CPN and the neural network without it. Recall the tighter the range of the eigenvalues is, the larger steps the gradient descent algorithm can take without worrying about divergence as explained in section $1.2$.

The graph above shows a kernel density estimation done on the input to hidden and hidden to output Hessian's at the near critical point. There are both negative and positive eigenvalues, especially in the hidden to output weights, therefore it is safe enough to say that we are at a saddle point (Turlach, 1993). The first graph represents the CPN neural network while next graph represents a non-normalized neural network. The CPN network shows a tighter distribution as well as more of the eigenvalues being focused near 0.

### 4.3 Toy Example

To ensure that the normalization is actually repelling the optimization point from saddle points, and that the results achieved in the experimental section are not due to some confounding factors we utilize a low-dimensional experiment to show the repelling effects of CPN.

We utilize the monkey saddle as the optimization surface. The monkey saddle has a saddle point surrounded by plateaus in all directions. It is defined as $x^3 - 3xy^2$. Referring to section $1.2$, we discussed that the direction of gradient descent algorithms was not the shortcoming of gradient descent algorithms around saddle points, but rather the problem was with the step size. What CPN should in theory do is allow the optimization routine to take larger steps.

Table 1: Hyper-parameters for Toy-Problem

| | | Algorithm | | | | CPN | | |
|---|---|---|---|---|---|---|---|---|
| | LR | Momentum | $\rho$ | $\beta_1$ | $\beta_2$ | $\alpha$ | $\beta$ | $\lambda$ |
| SGD | 0.01 | 0 | NA | NA | NA | 0.1 | 1.0 | 0.1 |
| SGD Accelerated | 0.01 | 0.9 | NA | NA | NA | 0.1 | 1.0 | 0.1 |
| AdaGrad | 0.01 | NA | NA | NA | NA | .5 | 1.0 | 0.001 |
| AdaDelta | 1.00 | NA | 0.95 | NA | NA | .5 | 1.0 | 0.001 |
| Adam | 0.01 | NA | NA | 0.9 | 0.999 | .5 | 1.0 | 0.001 |

Table 2: Final Loss After 120 Iterations for Toy-Problem

| | Non-CPN | CPN |
|---|---|---|
| SGD | $-2.00428E-12$ | $-8.192E9$ |
| SGD Accelerated | $-2.04018E-12$ | $-8.192E9$ |
| AdaGrad | $-1.75024E-11$ | $-0.00463$ |
| AdaDelta | $-2.46194E-12$ | $-2.22216$ |
| Adam | $-12.8413$ | $-12.9671$ |

Below are two figures. The first one shows the behavior of a five common gradient descent algorithms starting at a point near the saddle point (point: $(x = 0.0001, y = -0.0001)$) (Zeiler, 2012), (Duchi et al., 2010). The next figure shows the same algorithms starting at the same point but utilizing CPN. All visualization were done using the matplotlib library (Hunter, 2007).

The hyper-parameters used, were all the default hyper-parameters in the keras library apart from Adam (to make it visible on the graphs). All hyper-parameters are available in Table 1. SGD Accelerated refers to the standard SGD algorithm using momentum and nesterov acceleration. The CPN parameters were chosen using a very small discrete grid-search. In reality just about any reasonable arbitrary parameters can be chosen in order for CPN to work in this experiment, a grid-search was not neccessary to find a solution. This is why we reuse two sets of hyper-parameters for this toy problem. (Nesterov, 1983).

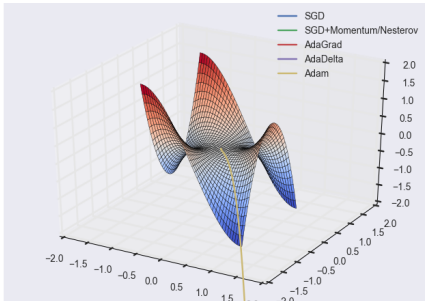

Figure 2: Non-CPN Optimization Paths

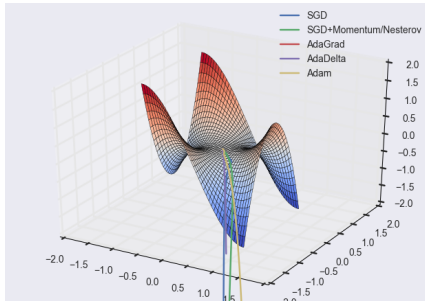

Figure 3: CPN Optimization Paths

Each algorithm performed better when coupled with CPN than without, the loss was computed using the monkey saddle equation above. All the losses for both CPN and Non-CPN are available in Table 2. CPN allowed the optimization algorithms to escape the saddle point quickly even though the gradient near the starting point of the optimization was near zero.

- Without CPN only the Adam algorithm escaped the plateau in less than a 1000 iterations.
- With CPN every algorithm apart from AdaGrad successfully escaped the plateau in less than 120 iterations, most notable being SGD Accelerated, which escaped in just 8 iterations.

From this toy example we can conclude that CPN does in fact repel the optimization algorithm away from saddle points, and therefore the results from the experiments are due to this phenomena and most likely no other confounding factors.

## 4.4 PERIODICITY AND TERMINAL BEHAVIOR

As shown in the experiments done on the CIFAR datasets, CPN has a tendency to force the optimization algorithm to act more chaotically. The exponential term in the normalization term is there to ensure that the optimization algorithm does not get stuck in an periodic path. It is trivial to see that as the time of the optimization goes toward infinity the impact of the normalization will tend toward 0. Therefore if the optimization algorithm does not reach a local minimum, but is rather in an elliptical path, assuming that the $\lambda$ term is not great enough to push the point out of the local minimum, the optimization algorithm will eventually reach the local minimum.

## 5 NOTES ON HYPER-PARAMETERS

Due to restrictions on our hardware resources, we did not have enough time to run a comprehensive study on the behavior of CPN with respect to its hyper-parameters. Throughout this paper the selection of hyper-parameters was kept rather simple. We selecting the hyper-parameters in a feasible range, and then adjusted them by either by hand around 4-8 times, or similarly by running a basic discrete grid-search that ran over that same amount of hyper-parameters. So in no way are the hyper-parameters for CPN chosen in this paper optimal for the various setups explained, but yet the results we found were somewhat substantial, which we find quite optimistic.

## 6 WEAKNESSES

- CPN with a exponential moving average for the $\phi$ function, introduces 2 extra hyper-parameters, not including the normalization scalar $\lambda$.
- In terms of implementation; CPN doubles the amount of memory needed for the optimization problem, as a trailing copy of the parameters must be kept.
- The fraction term in CPN will generally contain small floating points in both numerator and denominator and this can sometimes lead to numerical instability.
- If saddle points are reached at a really late time in the optimization algorithm, the exponential decay will nullify the effects of CPN. A possible solution would be to substitute the exponential decay term with some type of periodic decay.

## 7 CONCLUSION

In this paper we introduced a new type of dynamic normalization that allows gradient based optimization algorithms to escape saddle points. We showed empirical results on standard data-sets, that show CPN successful escapes saddle points on various neural network architectures. We discussed the theoretical properties of first order gradient descent algorithms around saddle points as well as discussed the influence of the largest eigenvalue on the step taken. Empirical results were shown that confirmed the hunch that the hessian of charged point normalized neural networks contains eigenvalues which are less in magnitude than their non-normalized counterpart.

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
