# Peer review of "Charged Point Normalization: An Efficient Solution to the Saddle Point Problem"

_ICLR 2017 — rejected_

[Official Review · AnonReviewer3 · rating 4 · confidence 3 · 15 Dec 2016]
**Momentum**

The research direction taken by this paper is of great interest. 
However, the empirical results are not great enough to pay for the weaknesses of the proposed approach (see Section 6). 
"Throughout this paper the selection of hyper-parameters was kept rather simple." but the momentum term of CPN is set to 0.95 
and not 0.9 as in all/most optimizers CPN is compared to. I suppose that the positive effect of CPN (if any) is mostly due to its momentum term.

[Official Review · AnonReviewer2 · rating 5 · confidence 4 · 20 Dec 2016]
**Interesting idea but needs more work**

Summary:
This paper proposes a regularizer that is claimed to help escaping from the saddle points. The method is inspired from physics, such that thinking of the optimization process is moving a positively charged particle would over the error surface which would be pushed away from saddle points due to the saddle point being positively changed as well. Authors of the paper show results over several different datasets.

Overview of the Review:
    Pros:
        - The idea is very interesting.
        - The diverse set of results on different datasets.
    Cons:
        - The justification is not strong enough.
        - The paper is not well-written.
        - Experiments are not convincing enough.

Criticisms:

I liked the idea and the intuitions coming from the paper. However, I think this paper is not written well. There are some variables introduced in the paper and not explained good-enough, for example in 2.3, the authors start to talk about p without introducing and defining it properly. The only other place it appears before is Equation 6. The Equations need some work as well, some work is needed in terms of improving the flow of the paper, e.g., introducing all the variables properly before using them.

Equation 6 appears without a proper explanation and justification. It is necessary to explain it what it means properly since I think this is one of the most important equation in this paper. More analysis on what it means in terms of optimization point of view would also be appreciated.

$\phi$ is not a parameter, it is a function which has its own hyper-parameter $\alpha$. 

It would be interesting to report validation or test results on a few tasks as well. Since this method introduced as an additional cost function, its effect on the validation/test results would be interesting as well.
The authors should discuss more on how they choose the hyper-parameters of their models. 

The Figure 2 and 3 does not add too much to the paper and they are very difficult to understand or draw any conclusions from. 

There are lots of Figures under 3.4.2 without any labels of captions. Some of them are really small and difficult to understand since the labels on the figures appear very small and somewhat unreadable.


A small question:

* Do you also backpropagate through $\tilde{\mW}_i^t$?

[Official Review · AnonReviewer1 · rating 4 · confidence 4 · 20 Dec 2016]
**Interesting idea, needs more rigorous comparison**

This paper proposes a novel method for accelerating optimization near saddle points. The basic idea is to repel the current parameter vector from a running average of recent parameter values. This method is shown to optimize faster than a variety of other methods in a variety of datasets and architectures.

On the surface, the proposed method seems extremely close to momentum. It would be very valuable to think of a clear diagram illustrating how it differs from momentum and why it might be better near a saddle point. The illustration of better convergence on the toy saddle example is not what I mean here—optimization speed comparisons are always difficult due to the many details and hyper parameters involved, so seeing it work faster in one specific application is not as useful as a conceptual diagram which shows a critical case where CPN will behave differently from—and clearly qualitatively better than—momentum.

Another way of getting at the relationship to momentum would be to try to find a form for R_t(f) that yields the exact momentum update. You could then compare this with the R_t(f) used in CPN.

The overly general notation $\phi(W,W)$ etc should be dropped—Eqn 8 is enough.

The theoretical results (Eqn 1 and Thm 1) should be removed, they are irrelevant until the joint density can be specified.

Experimentally, it would be valuable to standardize the results to allow comparison to other methods. For instance, recreating Figure 4 of Dauphin et al, but engaging the CPN method rather than SFN, would clearly demonstrate that CPN can escape something that momentum cannot.

I think the idea here is potentially very valuable, but needs more rigorous comparison and a clear relation to momentum and other work.

[Final Decision · Program Chairs · 06 Feb 2017]
**ICLR committee final decision**

The paper proposes a method for accelerating optimization near saddle points when training deep neural networks. The idea is to repel the current parameter vector from a running average of recent parameter values. The proposed method is shown to optimize faster than a variety of other methods in a variety of datasets and architectures.
 
 The author presents a fresh idea in the area of stochastic optimization for deep neural networks. However the paper doesn't quite appear to be above the Accept bar, due to remaining doubts about the thoroughness of the experiments.We therefore invite this paper for presentation at the Workshop track.